



# Turbulence structures and entrainment length scales in large offshore wind farms

Abdul Haseeb Syed[1], Jakob Mann[1], Andreas Platis[2], and Jens Bange[2]

[1]Department of Wind and Energy Systems, Technical University of Denmark, 4000 Roskilde, Denmark
[2]Environmental Physics, Geo- and Environmental Center, Eberhard Karls University of Tübingen, 72076 Tübingen, Germany

**Correspondence:** Abdul Haseeb Syed (absy@dtu.dk)

**Abstract.** The flow inside and around large offshore wind farms can range from smaller structures associated with the mechanical turbulence generated by wind turbines to larger structures indicative of the mesoscale flow. In this study, we explore the variation in turbulence structures and dominant scales of vertical entrainment above large offshore wind farms located in the North Sea, using data obtained from a research aircraft. The aircraft was flown upstream, downstream, and above wind farm clusters. Under neutrally stratified conditions, there is high ambient turbulence in the atmosphere and an elevated energy dissipation rate compared to stable conditions. The intensity of small-scale turbulence structures is increased above and downstream of the wind farm, and it prevails the mesoscale fluctuations. But in stable stratification, mesoscale flow structures are not only dominant upstream of the wind farm but also downstream. We observed that the vertical flux of horizontal momentum is the main source of energy recovery in large offshore wind farms, and it strongly depends on the magnitude of the length scales of the vertical wind velocity component. The dominant length scales of entrainment range from 20 to $\sim 60$ m above the wind farm in all stratification strengths, and in the wake flow these scales range from 10 m to $\sim 100$ m only under near-neutral stratification. For strongly stable conditions, negligible vertical entrainment of momentum was observed even just 2 km downstream of large wind farms. We also observed that there is a significant lateral momentum flux above the offshore wind farms, especially under strongly stable conditions, which suggests that these wind farms do not satisfy the conditions of an "infinite wind farm".

## 1 Introduction

The flow inside and around large wind farms is characterized by a wide range of spatio-temporal turbulence structures. The flow structures are not only influenced by the mechanical turbulence generated by wind turbines but also by the ambient turbulence present in the atmosphere (Meyers and Meneveau, 2013). Many numerical and analytical studies have been performed to understand the interactions between wind farms and atmospheric flow e.g. Porté-Agel et al. (2020) and Stevens and Meneveau (2017). Liu et al. (2018) suggested from their experiment inside a wind tunnel that integral time scales in the wind flow are decreased significantly above their modeled wind farm due to the development of an internal boundary layer and increase in turbulence above the wind farm. The atmospheric stratification also plays a significant role in the development of internal boundary layers (Savelyev and Taylor, 2005), and the evolution of turbulence structures downstream of large wind farms. Wu



and Porté-Agel (2017) described the effects of different free atmospheric stratification strengths on the upstream blockage and downstream wake lengths for large hypothetical wind farms using large-eddy simulations (LES). Their results showed that wind farms experience an increased blockage effect during strong atmospheric stratifications, because of sub-critical flow induced by wind farms i.e. the inertial forces cannot overcome the gravity-induced forces leading to Froude number Fr<1. Much longer downstream wakes are also observed in observations and numerical simulations during strong stratifications because of lower ambient turbulence and the development of fully-developed flow in large wind farms (Platis et al., 2020). Understanding the variation and evolution in turbulence structures in offshore wind farms is critical for the evaluation of the power fluctuations and turbine component loads and for the determination of optimal wind farm layouts.

In very large offshore wind farms, the kinetic energy entrainment from above the boundary layer is a primary source of energy replenishment. Ideally, this happens when a fully-developed flow is formed inside a wind farm i.e. when the flow becomes homogeneous in the streamwise direction and wind turbine wakes are fully merged (Emeis, 2013). Wu and Porté-Agel (2017) argued that the starting point of a fully developed region depends on the extent of thermal stratification: stronger stable stratification leads to the early development of a fully developed internal boundary layer inside large offshore wind farms. The fully developed region has been a point of interest lately since the height of many modern wind turbines often exceeds the atmospheric boundary layer (ABL) depth, especially during stable conditions in offshore sites. In real conditions, very few wind farms attain a fully-developed flow due to a number of reasons: the atmospheric conditions are not conducive for a fully developed internal boundary layer, the mean wind direction is not always aligned with the layout of wind turbines, or the wind turbine spacing is not constant causing heterogeneous flow conditions inside a wind farm. Moreover, recent LES studies have suggested that the distance required to attain a fully developed flow from the leading edge of a wind farm lies in the range of two orders of magnitude and larger than the ABL height (Wu and Porté-Agel, 2017) which is usually not attainable during weak thermal stratification.

Nonetheless, the vertical entrainment of energy or momentum is still a major source of energy recovery in the downstream direction of wind turbines and it has a strong dependence on atmospheric stratification (Abkar and Porté-Agel, 2013). It was observed Cortina et al. (2016) that vertical entrainment of mean kinetic energy (MKE) is more dominant during convective conditions while horizontal mixing or advection is more pronounced during stable atmospheric conditions. For a finite-size wind farm, where the flow regime does not enter into the fully developed flow, the kinetic energy distribution depends on the alignment configuration and spacing between wind turbines. Cortina et al. (2020) noted that under neutral atmospheric conditions, flow in the first few rows of the wind farm flow is energized by the advection of mean wind flow, while in the back rows energy entrainment from above is more responsible for the flow replenishment.

Most of the studies on entrainment are performed using LES on ideal wind farm layouts which do not truly depict the reality. Andersen et al. (2017) discussed the dominant length scales responsible for entrainment and their dependence on the streamwise spacing between wind turbines. Some LES studies and wind tunnel experiments have been utilized to develop analytical models for turbulent momentum fluxes above the wind farm sublayer (Markfort et al., 2018; Ge et al., 2021). These models are developed on the basis of top-down analytical models where the whole wind farm is considered as one roughness element ignoring the effect of multiple wakes superposed on each other. Hamilton et al. (2012) utilized the spectral analysis of





wind speed components measured in a wind tunnel experiment of a modeled wind farm to determine the dominant scales of entrainment. While these studies provide information about turbulence statistics and momentum fluxes above wind farms for simple layouts in ideal atmospheric conditions, there is an absence of such analysis in the literature that employs actual in-situ measurements on real wind farms.

Therefore, we evaluate in this study the dominant entrainment length scales and turbulence statistics around large offshore wind farms located in German Bight in the North Sea using in-situ measurements. The data was measured using the Dornier Do-128 research aircraft operated by TU Braunschweig as a part of a German research project called the Wind Park Far Field (WIPAFF) experiment. Detailed information about the flights and recorded data is described in Platis et al. (2018, 2020). The airborne data set of the WIPAFF project is accessible to the community via the PANGAEA database (Bärfuss et al., 2019). This study has the following objectives:

1. Evaluate the variation of turbulence length scales and the rate of energy dissipation upstream, above, and downstream of the offshore wind farms;

2. Investigate the effect of atmospheric stratification on turbulence length scales and energy dissipation rate in large offshore wind farms;

3. Investigate the variation of turbulent momentum fluxes around large wind farms;

4. Identify the dominant scales of entrainment around large offshore wind farms under different stratifications.

This article is organized in the following sections: (2) Data description and processing, in which details about the flights and the relevant data processing techniques are mentioned; important results are elucidated and discussed in (3) Results and (4) Discussion sections respectively, and finally the important conclusions of this study are presented in the last section (5) Conclusions.

## 2 Data description and processing

A total of 41 flights were conducted over the German Bight area in the North Sea from Sep 2016 to Oct 2017 as a part of the WIPAFF project. These flights are the first in-situ measurements of the far wake behind large offshore wind farm clusters. Some of these flights also recorded data upstream and above the wind farms. Several atmospheric parameters such as 3-dimensional wind vector, air temperature, pressure, and water vapor were logged using special instrumentation mounted on the Do-128 aircraft. The true airspeed of the aircraft was $66~\mathrm{ms}^{-1}$, and the sampling frequency of measurements was 100 Hz (Platis et al., 2018).

Six flights out of the total 41 were suited for our analysis which were operated above two different wind farm clusters as described in Table 1 and Table 2. The North wind farm cluster comprises three wind farms namely: AW (Amrumbank West), NO (Nordsee Ost), and MW (Meerwind Süd). The South wind farm cluster comprises two wind farms called Godewind I and II respectively (see Figure 1). Detailed information about the turbine types in these wind farms and their technical specification





**Table 1.** Details of the four flights operated upstream, above, and downstream of offshore wind farms analyzed in this study. The flight numbers represent the numbers given in the WIPAFF campaign. The abbreviations used for wind farms are: AW (Amrumbank West), NO (Nordsee Ost), MW (Meerwind Süd) and GW (Godewind).

| Flight No. | Flight Date | Time (UTC) | Wind Farms | Altitude (AMSL)(m) | Mean Wind Speed U (m s$^{-1}$) | Mean Wind Direction (°) | Lapse Rate($\gamma$) (K (100 m)$^{-1}$) |
|---|---|---|---|---|---|---|---|
| 32 | Aug 09, 2017 | 0834-1236 | AW, NO, MW | 200 | 15.9 | 215 | 0.24 |
| 33 | Aug 09, 2017 | 1309-1705 | AW, NO, MW | 200 | 12.9 | 240 | 0.18 |
| 39 | Oct 14, 2017 | 1259-1640 | GW I, II | 250 | 15.3 | 250 | 0.91 |
| 40 | Oct 15, 2017 | 0706-1108 | GW I, II | 250 | 14.2 | 199 | 1.13 |

**Table 2.** Same as Table 1 but for the flights operated mainly downstream of the wind farm

| Flight No. | Flight Date | Time (UTC) | Wind Farms | Altitude (AMSL)(m) | Mean Wind Speed U (m s$^{-1}$) | Mean Wind Direction (°) | Lapse Rate($\gamma$) (K (100 m)$^{-1}$) |
|---|---|---|---|---|---|---|---|
| 7 | Sep 10, 2016 | 0733-1115 | AW, NO, MW | 100 | 8.5 | 191 | 0.18 |
| 30 | Aug 08, 2017 | 0834-1233 | AW, NO, MW | 100 | 7.6 | 85 | 0.23 |

can be found in Siedersleben et al. (2020). Table 1 consists of flights operated upstream, above, and downstream of the wind farms, while Table 2 consists of flights that have several legs in the downstream direction and one upstream leg. The flights mentioned in Table 1 are analyzed to study turbulence structures and momentum fluxes in Sections 3.1, 3.2, and 3.3 while the flights in Table 2 are chosen to study the variation in dominant length scales of entrainment at hub height in the wake of large wind farm cluster in Section 3.4.

Figure 1 illustrates the 4 flights mentioned in Table 1 where three distinct flight legs and the location of wind turbines are shown. Here we only analyzed the portion of the flight legs projected to the wind farm cluster in the mean wind direction. It can be seen from the Figure 1 that the upstream flight legs in Flight 32 and Flight 33 consist of undisturbed wind flow while in Flight 39 and Flight 40, a portion of upstream flight legs is carried above an upstream wind farm called Nordsee One which, as we will discuss later, disturb the incoming flow and add turbulence to it.

Vertical profiles were also measured in the vicinity of the wind farms for further information on the marine atmospheric boundary layer. These measurements were recorded as the aircraft changed its altitude from ∼50 m to ∼1000 m above mean sea level (AMSL). The potential temperature profiles measured over this range of altitude during the four flights in Table 1 are shown in Figure 2. The potential temperature profiles on 9th August (Flight 32, and 33) suggest weak, almost neutral thermal stratification, while very stable conditions were prevalent during 14th and 15th October (Flight 39, and 40). The average



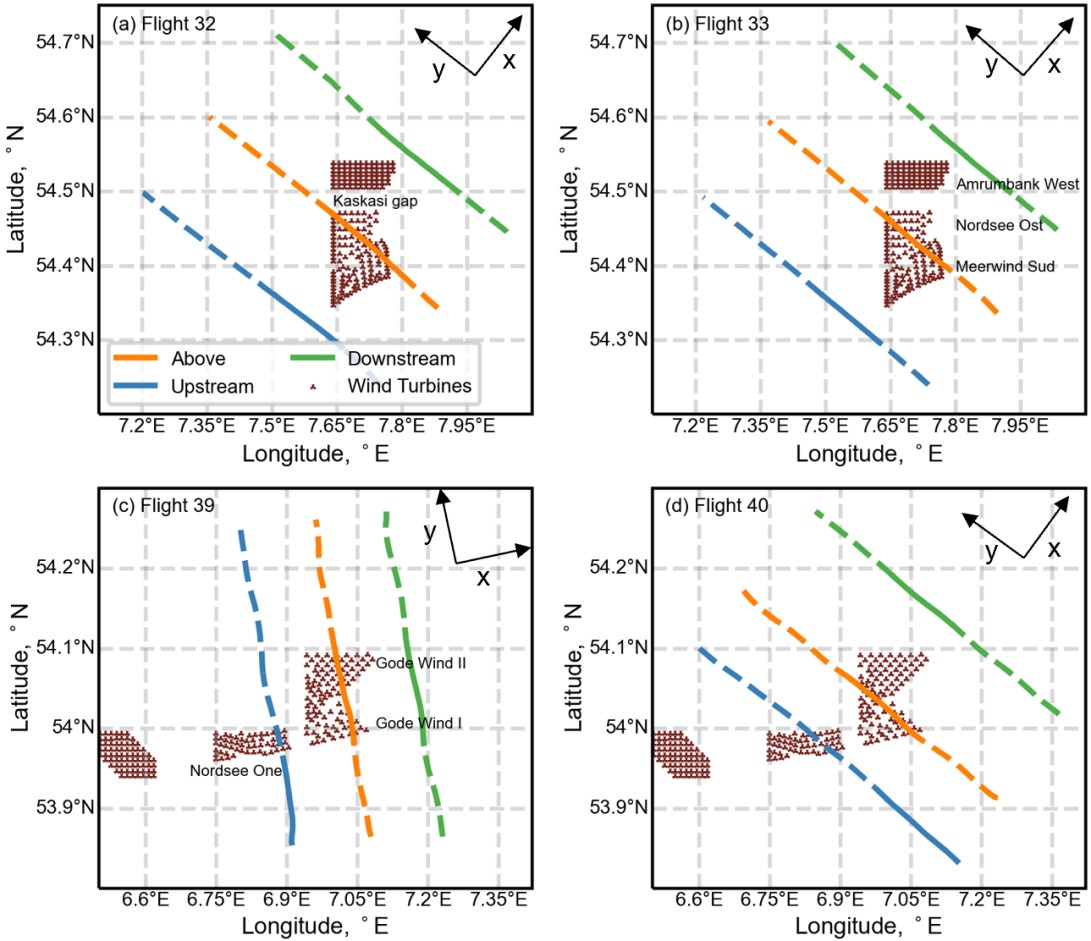

**Figure 1.** The four flights operated above two different wind farm clusters in the North Sea. (a) and (b) present the flight legs above Meerwind Süd and Nordsee Ost wind farms, (c) and (d) shows the flight legs above Godewind 1 and 2 wind farms. The x-y axis presents the coordinate system in which geographical wind vectors are rotated, where x is the mean wind direction and y is the transverse direction in which flight measurements were recorded. The portion of the flight legs represented by solid lines is chosen for the analysis presented in this study.

potential temperature gradient (also known as lapse rate, $\gamma$) was 0.24 and 0.18 K ($100\,\mathrm{m}^{-1}$) for Flights 32 and 33 respectively, while for Flights 39 and 40 it was 0.91 K ($100\,\mathrm{m}^{-1}$) and 1.13 K ($100\,\mathrm{m}^{-1}$) respectively. Moreover, the lapse rate is considered to be a robust criterion for atmospheric stability classification in German Bight by Platis et al. (2022). In Table 1 and Table 2 we have specified the lapse rate observed during all six flights between height intervals of 50 m and 100 m AMSL. The lapse rate can provide a good qualitative estimate of the thermal stratification and vertical mixing present in the atmosphere. As discussed in their study Platis et al. (2022) observed an inverse correlation of 68% between lapse rate and vertical velocity component variance $\langle w'w' \rangle$ during the 41 flights of WIPAFF campaign.





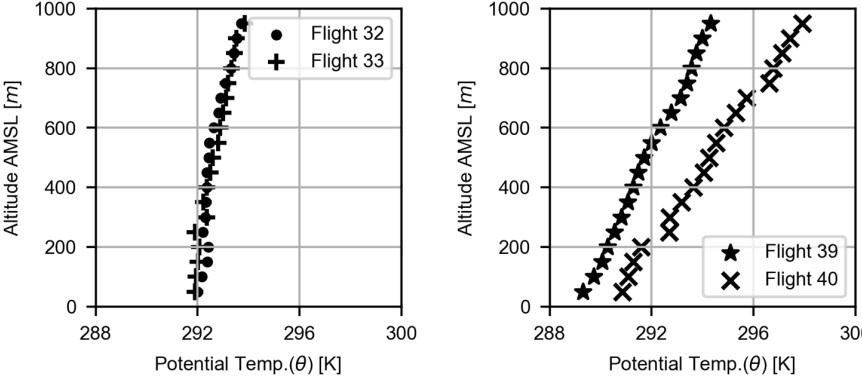

**Figure 2.** The vertical profiles of potential temperature measured by the aircraft during the four flights mentioned in Table 1. Each point represents an average of data points in a 50 m interval.

The wind components logged by the aircraft are first converted to the geographical coordinate system (Bange et al., 2013; Desjardins et al., 2021). For the analysis presented in this study, the geographical wind vectors are transformed into the right-handed coordinate system (see Figure 1) defined by the direction of mean wind using Equation 1:

$$\begin{bmatrix} u_t \\ v_t \end{bmatrix} = \begin{bmatrix} \cos\phi & \sin\phi \\ -\sin\phi & \cos\phi \end{bmatrix} \begin{bmatrix} u \\ v \end{bmatrix} \tag{1}$$

where $u$ and $v$ are the geographical horizontal wind components, positive in the East and North directions, respectively. The wind direction $\phi$ is given in the mathematical convention with zero degrees for westerly wind and 90 degrees for southerly wind.

Based on the values of the observed lapse rate during the flights, we classify the atmospheric conditions during Flight 32, and 33 as "weakly stratified" and Flight 39, and 40 as "strongly stratified". Similarly, the lapse rate values recorded during Flight 7 and 30 indicate the presence of "weakly stratified" or "near-neutral" atmospheric conditions (see Table 2). Figure 3 shows the variation in the transformed horizontal wind speed component $u_t$ over the duration of whole flight legs recorded upstream, above, and downstream of the wind farms. The left column (Figure 3 (a), (b), and (c)) shows the variation of $u_t$ during weak stratification case (Flight 32) while the right column (Figure 3 (d), (e), and (f)) represents the measurements obtained during strongly stable stratification (Flight 39). It can be distinctly observed that there are significant small-scale ambient turbulence structures present during weak stratification, both upstream and outside of the wind farm boundary. The turbulence generated by wind turbines is not clearly distinguishable because of the high ambient turbulence. We can also observe that the reduction in wind speed above and downstream of the wind farm is not remarkable during weak stratification. Conversely, there is very low small-scale ambient turbulence in the strong stable conditions, except a small portion in the upstream flight leg (Figure 3 (d)) caused by the presence of an upstream wind farm called Nordsee One (see Figure 1 (c)). The turbulence generated by the





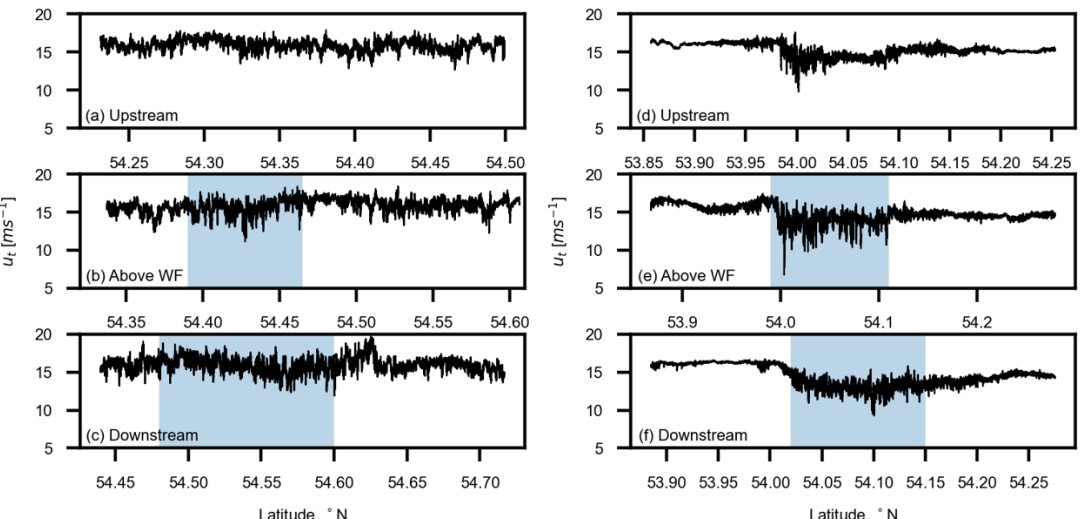

**Figure 3.** The horizontal wind speed component transformed in the mean wind direction for (a, b, and c) Flight 32 (Altitude: 200 m AMSL), weak stratification and (d, e, and f) Flight 39 (Altitude: 250 m AMSL), strong stratification. The blue shaded areas represent wind farm boundaries in (b) and (e), and the wind farm's wake region in the downstream direction in (c) and (f).

wind turbines is distinguishable and significant in this case, as is the reduction in wind speed above and downstream of the wind farm.

## 3 Results

### 3.1 Turbulence scales

The flight legs are oriented approximately orthogonal to the mean wind direction. To estimate the dominant turbulence length scales of the wind component in the mean wind direction, the integral length scale is used:

$$\rho_{u_t u_t}(\eta) = \frac{\overline{u_t'(y)u_t'(y+\eta)}}{\sigma_{u_t}^2} \tag{2}$$

where $\rho_{u_t u_t}(\eta)$ represents the auto-correlation function of $u_t$ in the direction perpendicular to the mean wind flow (along the orientation of the flight leg) denoted by $y$. $u_t'(y)$ corresponds to the fluctuations, $\eta$ is the space lag in $y$ direction and the variance of $u_t$ is denoted by $\sigma_{u_t}^2$.

The auto-correlation diagrams in Figure 4 help distinguish turbulent from mesoscale motions during different stratification strengths. For instance, Figure 4(a) and (b), the upstream flight legs in weakly stratified cases: here the turbulence causes a monotonically and steeply decreasing auto-correlation until a spatial lag $\eta$ of a few hundred meters. This represents high



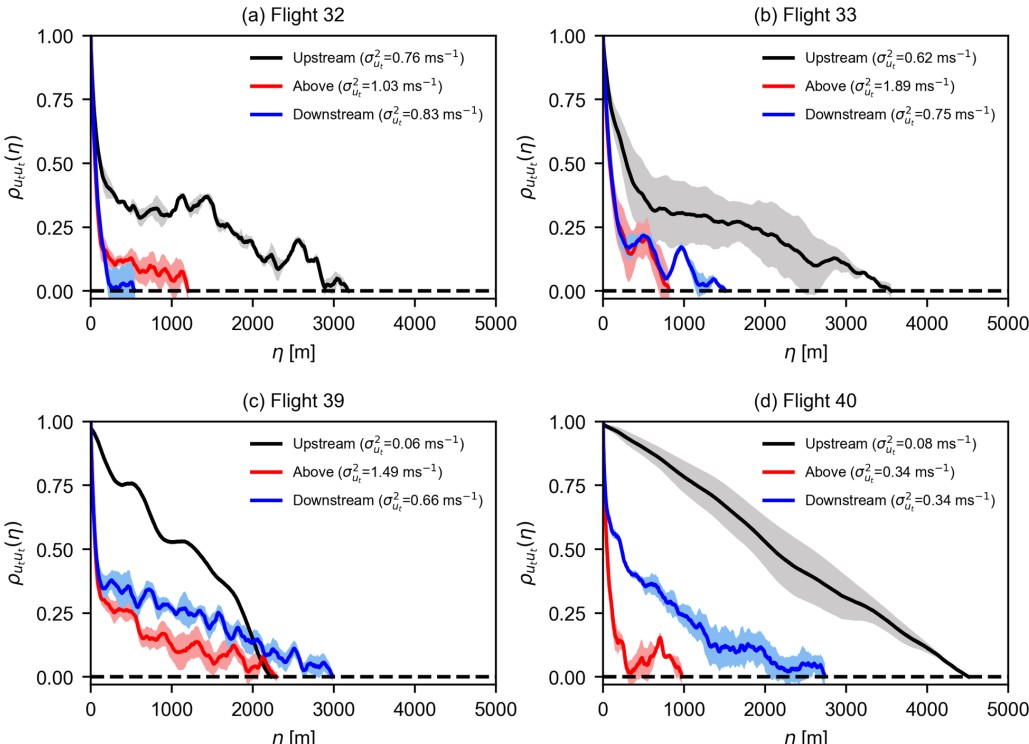

**Figure 4.** The autocorrelation of the along-wind component in the transverse direction plotted upstream, above, and downstream of the wind farms for the four flights mentioned in Table 1. The grey, red, and blue shaded areas represent the Standard Error of the Mean (SEM) due to averaging of data from multiple flight legs. Flight 32 and 33 represent weakly stratified atmospheric conditions, while Flight 39 and 40 were recorded when the atmosphere was strongly stratified.

ambient turbulence in the atmosphere due to increased vertical mixing. Then the auto-correlations is about constant and even increases at $\eta$ of $\sim$1 km (Figure 4(a)) probably indicating mesoscale structures which are not yet disturbed by the wind farm. The intensity of small-scale turbulence is increased above and downstream of the wind farm due to turbulence generated by wind turbines and it overshadows the mesoscale structures, also seen by a weak correlation $\rho_{u_t u_t}(\eta)$ at large spatial lags.

During strongly stable conditions illustrated in Figure 4 (c) and (d), the mesoscale fluctuations in the upstream flight legs are more dominant clearly shown by large values of $\rho_{u_t u_t}(\eta)$ at large spatial lags. Above the wind farm during strong stable stratification, there is a huge presence of small-scale turbulence and it is not much different from the weak stratification. Since there is not a lot of vertical mixing present due to the stable stratification, the downstream measurements (Figure 4(c) and (d)) suggest that fluctuations caused by the wind turbines are dominant in the wake flow of wind farms, but still we observed large values of $\rho_{u_t u_t}(\eta)$ at large spatial lags indicative of mesoscale structures. This can also be seen in Figure 3 (f) where small-scale turbulence starts to die out in the downstream flight leg. This indicates that the wakes created by wind turbines will

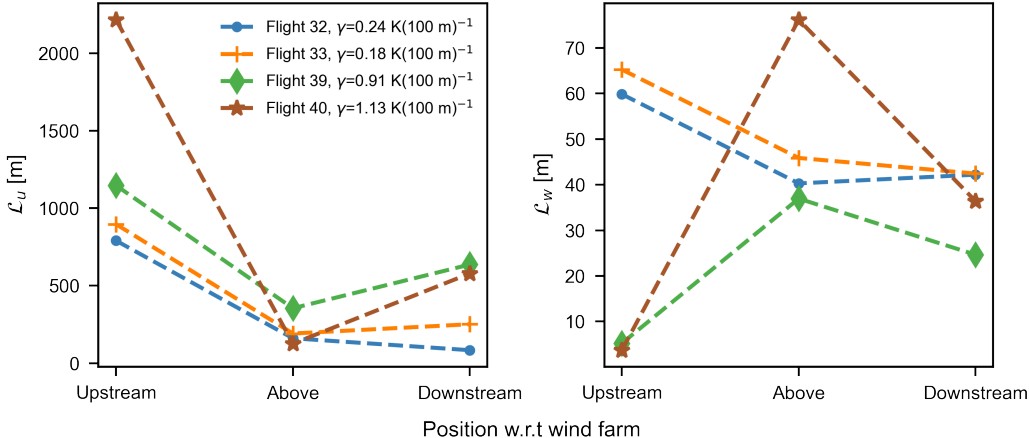

**Figure 5.** Longitudinal length scales $\mathcal{L}_u$, and vertical length scales $\mathcal{L}_w$ at different positions relative to the wind farm plotted for all the four flights.

last much longer in these conditions due to low ambient turbulence and the low intensity of small-scale structures (Platis et al., 2022).

The integral length scales $\mathcal{L}$ can be obtained by taking the integral of $\rho_{u_t u_t}(\eta)$ from 0 to the point of first zero crossing of $\rho_{u_t u_t}(\eta)$ i.e. $\eta_0$.

$$\mathcal{L}_u = \int_0^{\eta_0} \rho_{u_t u_t}(\eta) \, d\eta \tag{3}$$

The integral length scale $\mathcal{L}_u$ is much larger during strongly stable stratification in the upstream and downstream of the wind farms signifying larger length and time scales for the $u$-component. This is presented in Figure 5 (a) and (b) where longitudinal length scales $\mathcal{L}_u$, and vertical length scales $\mathcal{L}_w$ at different positions relative to the wind farm are plotted for all the four flights. The large values of $\mathcal{L}_u$ indicate the presence of 2-D turbulence where the vertical mixing is extremely low, and hence the lower values of $\mathcal{L}_w$ at corresponding positions. Due to increased vertical mixing above the wind farm, $\mathcal{L}_w$ increases significantly for strong stable conditions and then decreases in the downstream positions for all flights. For weak stratification, significant changes in $\mathcal{L}_w$ were not observed, although the slight drop in magnitude from upstream to above the wind farm positions can be referred to unsteady atmospheric conditions observed during the flight legs at the two locations.

## 3.2 The rate of energy dissipation

In this section, we discuss the rate of energy dissipation $\epsilon$ by plotting the compensated spectra for the flights mentioned in Table 1. The purpose of doing that is to evaluate at what rate the energy is being dissipated from large-scale eddies to smaller flow structures in either the ambient turbulence or in the turbulence generated by wind turbines. In the inertial subrange, the





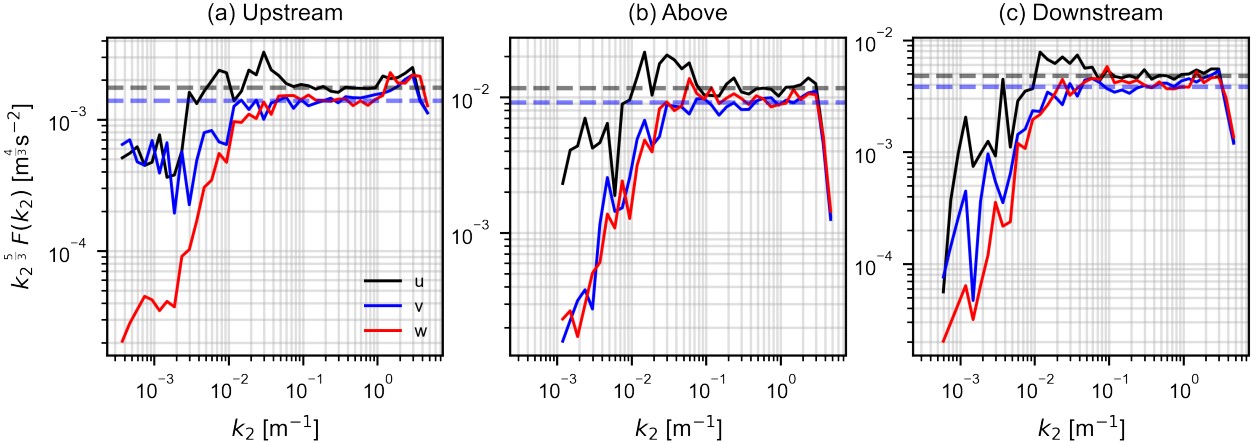

**Figure 6.** An illustration of the compensated spectra in terms of wave number $k_2$. This specific spectra represent Flight 33 (weak stability). The dashed black and blue lines represent the average values for $u$ and $v$ spectra, respectively, in the inertial subrange.

one-point, two-sided velocity spectra in terms of wavenumber $k_2$ ( where $k = 2\pi f / \overline{U}$; $f$ is the sampling frequency and $\overline{U}$ is the magnitude of the resultant vector of aircraft speed and incoming wind speed) are given by Equation 4 and Equation 5 (Mann, 175 1994).

For the $v$ wind component:

$$F_{22}(k_2) = \frac{9}{55} \alpha \epsilon^{\frac{2}{3}} k_2^{-\frac{5}{3}} \qquad (4)$$

For the $u$ and $w$ wind components:

$$F_{11}(k_2) = F_{33}(k_2) = \frac{12}{55} \alpha \epsilon^{\frac{2}{3}} k_2^{-\frac{5}{3}} \qquad (5)$$

which implies (Pope, 2000):

$$F_{11}(k_2) = F_{33}(k_2) = \frac{4}{3} F_{22}(k_2) \qquad (6)$$

where $\alpha$ is the spectral Kolmogorov constant having a value of ~1.7 (Mann, 1994), and $k_2$ is the wavenumber along the flight path and across the mean wind direction. Notice that as a function of $k_1$, which is the more usual case for, for example, anemometers mounted in meteorological masts, then $F_{22}(k_1) = F_{33}(k_1) = \frac{4}{3} F_{11}(k_1)$.

By plotting the spectra, we expect a constant value in the inertial subrange which can be used to identify the rate of energy dissipation $\epsilon$ from Equation 4 and Equation 5.

Figure 6 displays the compensated spectra for weak thermal stratification (Flight 33) in terms of wavenumber $k_2$ along the flight path, and the inertial subrange can be distinctly observed from wavenumbers between ~$10^{-1}$ m$^{-1}$ and ~$10^0$ m$^{-1}$. It is pertinent to point out the discrepancy found in the $w$-spectra which should be equal to $u$-spectra in the inertial subrange

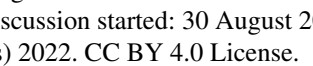



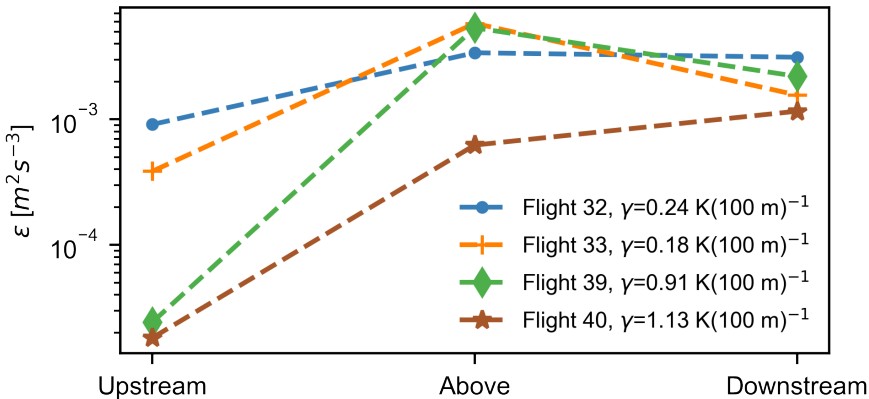

**Figure 7.** The rate of energy dissipation $\epsilon$ recorded upstream, above, and downstream of the wind farms in near-neutral and strongly stable conditions.

as given by Equation 5 (Saddoughi and Veeravalli, 1994). This deviation could possibly be linked to a calibration error in the vertical velocity component measured by the instruments installed on the aircraft. The average values for compensated $u$ and $v$ spectra in the inertial subrange are denoted by dashed black and blue lines, respectively, in the plot and can be used to evaluate the rate of energy dissipation using Equation 4 and Equation 5. A similar procedure was applied on all the four flights described in Table 1, and the mean values of the rate of energy dissipation $\epsilon$ obtained from $u$ and $v$ wind components are plotted

in Figure 7. From the plot, it can be observed that the upstream energy dissipation rate $\epsilon$ is much higher during near-neutral stratification (Flights 32, and 33), almost $\sim$40 times as compared to strongly stable stratification, and it corresponds with the high ambient turbulence during neutral stratification. Above the wind farm, there is not much difference in the dissipation rate $\epsilon$ between neutral and stable conditions as it mostly depends on the layout of the wind farm and the incoming wind speed. The lower value of $\epsilon$ in Flight 40 can be partially referred to as highly stable conditions and the location of the flight leg above the

wind farm caused it to not be exposed to a large number of wind turbines from the mean wind direction (see Figure 1 (d)). The dissipation rate $\epsilon$ in the downstream direction depends on a lot of factors: configuration and density of wind turbines in the cluster, upstream wind speed, wind direction, and distance of the downstream flight leg from the wind farm trailing edge. From Figure 7 it can be seen that the energy dissipation rate $\epsilon$ is significantly alike for all cases in downstream of wind farms. In all cases, $\epsilon$ remains almost similar or there is a drop in $\epsilon$ downstream of the wind farm, except for Flight 40 which is because

of the large number of wind turbines affecting the portion of the downstream flight leg as compared to the flight leg portion above the wind farm (see Figure 1 (d)).



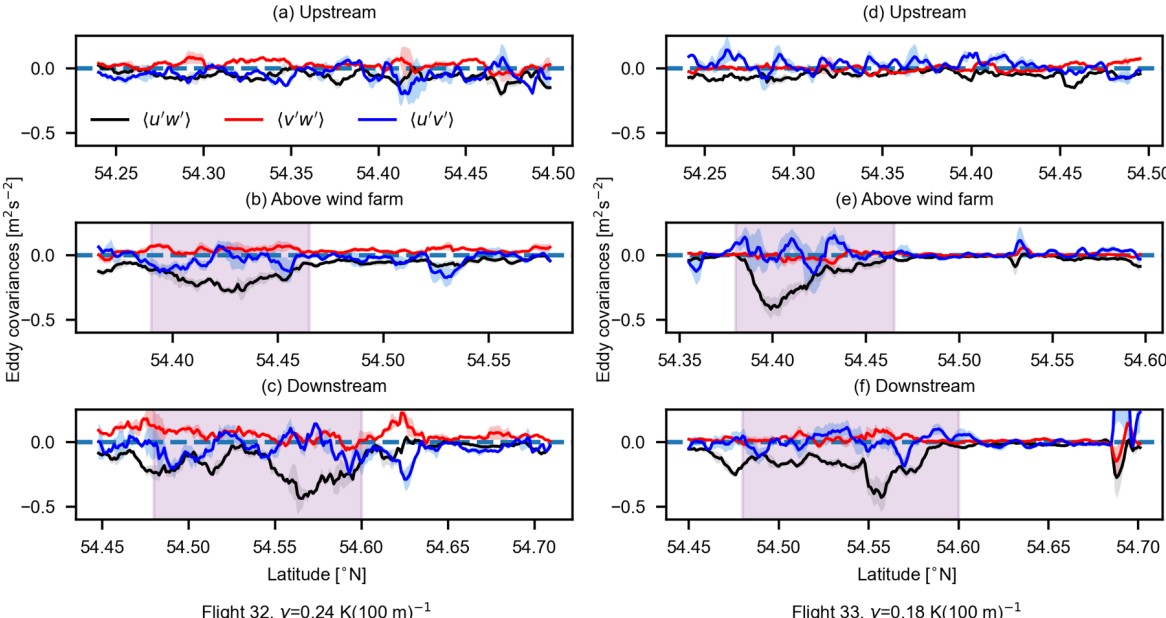

**Figure 8.** The variation in $\langle u'w' \rangle$, $\langle v'w' \rangle$, and $\langle u'v' \rangle$ measured upstream, above, and downstream of the wind farm for Flight 32 and Flight 33 (weak thermal stratification). The black, red, and blue shaded areas represent the Standard Error of Mean (SEM) due to averaging of data from multiple flight legs. The pink shaded areas on the abscissa represent wind farm boundary in (b) and the wind farm's wake projection in the downstream direction in (c).

### 3.3 Turbulent momentum fluxes

Here we analyze the top-down and lateral influx of momentum into the wind farms as the flow energy is depleted by the presence of wind turbines. This influx of momentum helps replenish the energy available to wind turbines and also recovers

the wind in the wake of wind farms.

Figure 8 displays the variation in the eddy covariances of velocity components: $\langle u'w' \rangle$, $\langle v'w' \rangle$, and $\langle u'v' \rangle$ for Flight 32 and 33 tracks recorded upstream, above, and downstream of the North wind farm cluster under weak thermal stratification. The Reynolds stress component $\langle u'w' \rangle$ (i.e. the vertical transport of horizontal momentum along the main wind direction) is largely responsible for the influx of momentum. From Figure 8, we can observe that there is already some momentum flux in upstream

of the wind farm in both flights. This is due to the presence of shear and lack of stratification, which implies large vertical velocity scales $\mathcal{L}_w$ in the atmosphere, which enhances the vertical mixing and thus the magnitude of $\langle u'w' \rangle$. Above the wind farm, we see a rise in $\langle u'w' \rangle$ due to the turbulence and shear generated by wind turbines, which increase vertical mixing and thus a downward flow of momentum. The large values of $\langle u'w' \rangle$ in the downstream flight legs (Figure 8 (c) and (f)) corresponds with the location of the wake. Here we observed two distinct peaks of $\langle u'w' \rangle$: the one at lower latitudes corresponds to the

downstream wakes of the two wind farms below the Kaskasi gap and the large peak at higher latitudes refers to the downstream



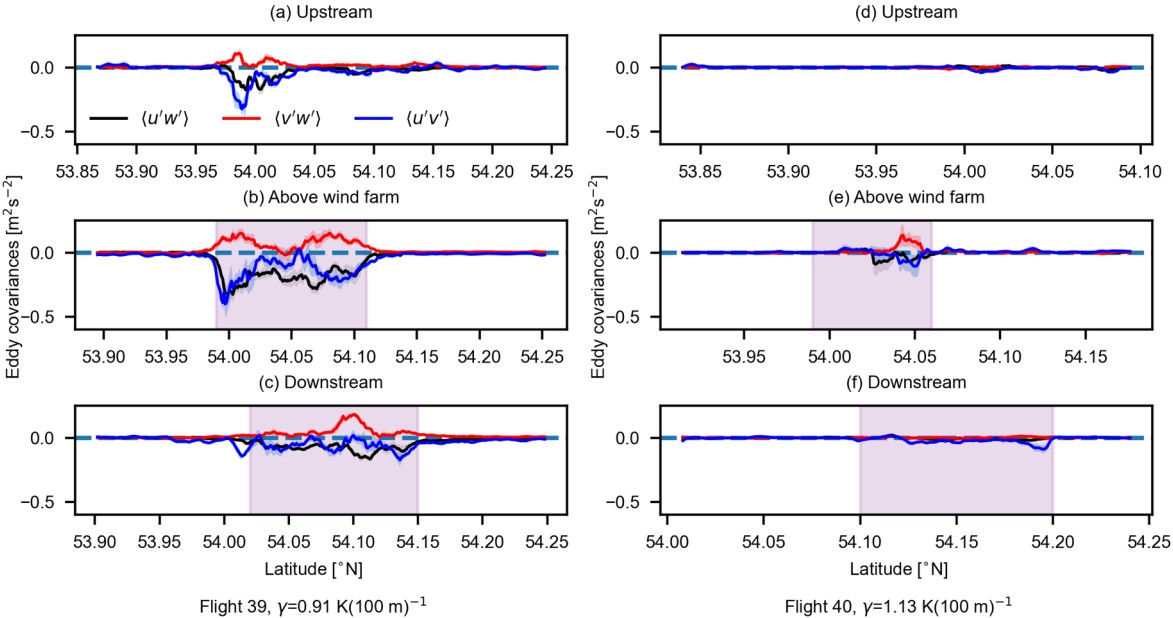

Flight 39, $\gamma$=0.91 K(100 m)$^{-1}$      Flight 40, $\gamma$=1.13 K(100 m)$^{-1}$

**Figure 9.** Same as Figure 8, but for Flight 39 and Flight 40 (strong thermal stratification)

wake of the dense Amrumbank West wind farm. Non-zero $\langle u'w' \rangle$ momentum flux was also observed outside the wind farm boundary in Figure 8 (b), and (e) which is an indication of high ambient turbulence. The variation in $\langle v'w' \rangle$ does not differ significantly in the three positions relative to the wind farm, and its magnitude is considerably lower as compared to $\langle u'w' \rangle$ above the wind farm as well. For the lateral momentum flux component $\langle u'v' \rangle$, we observed no distinct pattern due to the

225 presence of wind farms. Rather the variation in $\langle u'v' \rangle$ seemed much more chaotic in all three positions. This can be attributed to the fact that we do not observe a considerable reduction in longitudinal wind component $u$ under weak stratification above and downstream of the wind farm (also see Figure 3) due to high ambient turbulence in the atmosphere. Thus a lack of sharp gradient of $u$ in the transverse direction corresponds to no significant lateral momentum flux $\langle u'v' \rangle$.

The variation of momentum flux components during strong thermal stratification is shown in Figure 9. The change in eddy

covariances of the three velocity components is displayed with respect to the flight legs taken during Flight 39 and Flight 40. During strong stable stratification, there is no top-down flow of momentum $\langle u'w' \rangle$ outside the wind farm boundary (the shaded pink area in Figure 9). A small peak in $\langle u'w' \rangle$ during the upstream flight legs was observed due to the presence of Nordsee One wind farm below the upstream flight leg (see Figure 1). While there is a strong momentum flux above the wind farm, it decreases significantly in downstream of the wind farm resulting from the lack of vertical mixing i.e. small vertical length scales

$\mathcal{L}_w$ in the atmosphere. For the vertical flux of lateral momentum $\langle v'w' \rangle$, the non-zero values are also observed only due to the disturbance in flow generated by wind farms. Furthermore, the values observed for the vertical flux of lateral momentum $\langle v'w' \rangle$ are less than half of the vertical flux of horizontal momentum $\langle u'w' \rangle$. Due to large reduction in the longitudinal wind speed





component $u_t$ during strong stable conditions (see Figure 3), we also observe a considerable lateral momentum flux component $\langle u'v' \rangle$ in all the three positions for Flight 39. Some more discussion on the lateral momentum flux $\langle u'v' \rangle$ is presented in the

next section. The two peaks in the value of eddy covariances for the flight legs recorded during Flight 39 above the wind farm (see Figure 9 (b)) are considered to be a result of the layout of the wind turbines in Godewind I and II wind farms and the location of the flight leg. The magnitudes of all the three eddy covariances during Flight 40 are much smaller due to stronger thermal stratification and the turbulent fluxes are almost negligible even in the downstream wake of the wind farms.

### 3.4  Dominant scales of entrainment

In this section, we analyze the length scales responsible for the vertical entrainment of momentum in large wind farms. We use the data from the flights mentioned in Table 1 and Table 2 representing different levels of stratification based on the observed lapse rate $\gamma$. Here we only evaluate the dominant scales of $\langle u'w' \rangle$ since it is the most dominant form of entrainment compared to $\langle u'v' \rangle$ in large offshore wind farms. After selecting the part of the flight legs projected downstream of the wind farm in the mean wind direction, we look at the $uw$-cross-spectrum to determine the dominant length scales. This is done for both wake

flow and undisturbed flow for all flights. The relation used to find most dominant scales of vertical entrainment is defined by k, where:

$$ln\left(\frac{k}{k_0}\right) = \frac{\int_0^\infty \mathbb{R}(F_{uw}(k_2))\ln(\frac{k_2}{k_0})dk_2}{\int_0^\infty \mathbb{R}(F_{uw}(k_2))dk_2} \tag{7}$$

and

$$k = k_0 \exp\left(ln\left(\frac{k}{k_0}\right)\right) \tag{8}$$

Here $k$ is the dominant wave number in $\text{m}^{-1}$, $\mathbb{R}(F_{uw}(k_2))$ is the real part of $uw$-cross spectrum as a function of wavenumber $k_2$, and $k_0$ is a reference wave number. The dominant wavenumber can be understood as the center of gravity of the pre-multiplied spectrum plotted on a logarithmic $k_2$-axis. A length scale $1/k$ is defined in this way.

### 3.4.1  Close to the wind farm: Upstream, Above, and Downstream positions

The length scales contributing to the wake recovery and the downward momentum flux during Flights 32, 33, 39, and 40 are

shown in Figure 10. The impact of thermal stratification can be observed in the dominant length scales observed during the four flights. During Flights 32 and 33, when the thermal stratification is quite low signifying the presence of large vertical mixing, we observed relatively larger length scales contributing to the vertical entrainment in both wake and undisturbed flow. These scales range from $\sim$40 m to $\sim$100 m, indicative of strong turbulence and vertical mixing in the flow. For Flights 39 and 40, we did not observe any entrainment of momentum flux in the upstream position hence the contributing length scales are not

presented. Similarly, negligible momentum flux was observed downstream of the wind farm clusters during Flight 40 due to strongly stable conditions, hence no data point is presented. For strongly stable conditions, we observed similar entrainment-contributing length scales above the wind farm as near-neutral conditions, ranging from $\sim$20 m to $\sim$40 m. In the downstream of

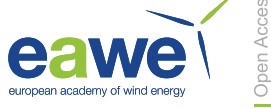
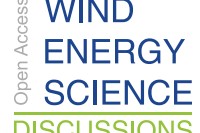


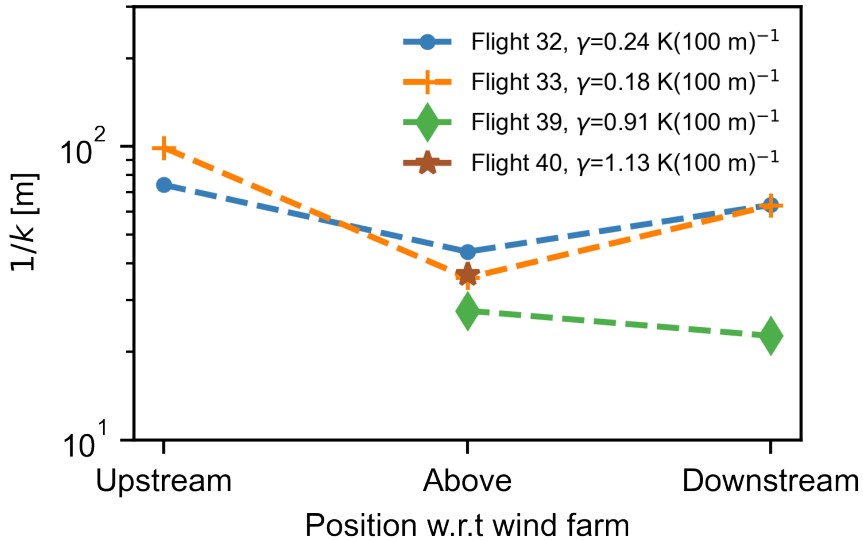

**Figure 10.** Dominant entrainment length scales of $\langle u'w' \rangle$ at different positions around the wind farms clusters. These measurements were recorded at about 60 m above the rotor top tip in these wind farms. Note that there are no upstream data points for Flight 39, and no upstream and downstream points for Flight 40 due to negligible momentum entrainment at these positions.

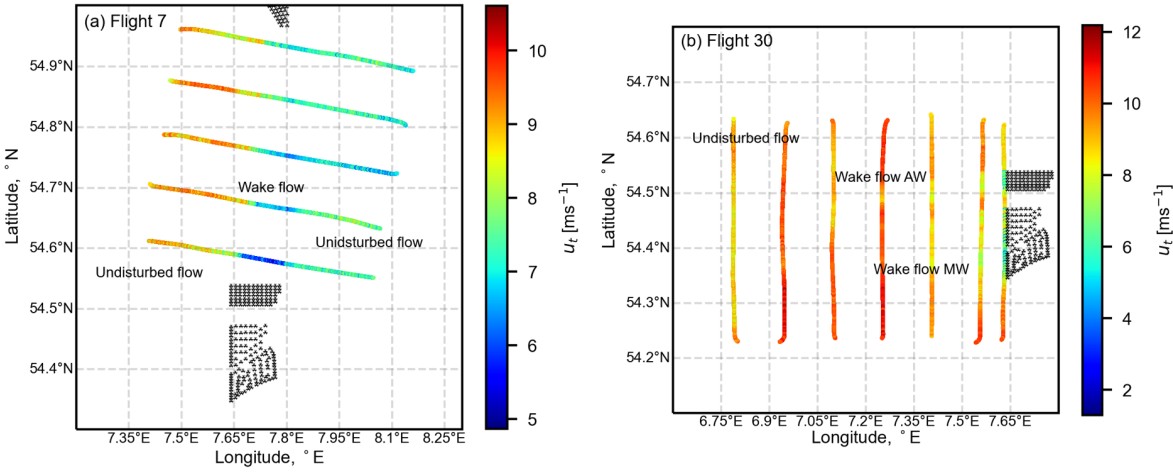

**Figure 11.** Downstream flight legs recorded during (a) Flight 7 and (b) Flight 30. The color bar shows variation in the horizontal wind speed component $u_t$. The wake flow and undisturbed flow is annotated for both flights.

wind farms during strongly stable conditions (Flight 39), a slight decrease in dominant length scales of entrainment is observed. This is because of the prevalent strongly stable conditions which inhibit vertical mixing.



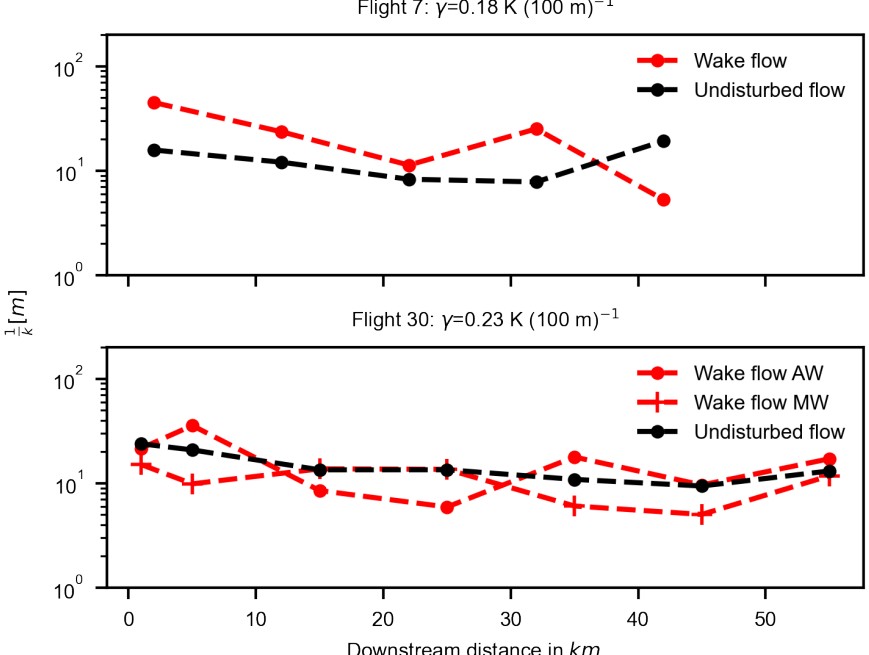

**Figure 12.** Dominant entrainment length scales in the downstream of the wind farms cluster. These measurements were recorded at hub-height level i.e. 100m AMSL. Note that only 5 downstream flight legs were recorded for Flight 7, and seven flight legs for Flight 30.

### 3.4.2 Far wake flow field

The dominant length scales of entrainment further downstream of the wind farm clusters are also analyzed to see their effect on wake recovery. The flights used for this purpose are detailed in Table 2. The two flights were recorded in downstream of the North wind farm cluster (AW, NO, MW). Flights 7 consists of wind approaching the wind farms from a South-East direction thus merging the wakes of all three wind farms into a single wake. An illustration of the wind speed deficit observed during the flight legs recorded during Flight 7 is shown in Figure 11 (a), where the wake flow can be distinctly observed in the downstream direction. Flight 30 consists of wind approaching from the East direction, thus two separate wakes from AW wind farm, and NO, MW wind farms are observed (see Figure 11 (b)). The undisturbed flow during both flights is also specified in the illustration.

During Flight 7, when the thermal stratification is quite weak signifying the presence of high vertical mixing, the dominant length scales of entrainment range from $\sim$10 m to $\sim$60 m, indicative of strong turbulence in the flow. An interesting observation during Flight 7 is the presence of large-scale structures in the wake flow (almost 3-5 times larger) as compared to the undisturbed flow. This is a consequence of shear-induced vertical mixing generated by the wind turbines which enhances the entrainment process. Flight 30 represents a flight when the predominant wind direction is East i.e. coming directly from the





land, as seen in Figure 11 (b). This causes two separate distinguishable wake flows as seen in the illustration (see Figure 12).
The entrainment length scales range from <10 m to ∼40 m because of the weak thermal stratification, for both wake flows and undisturbed flow. Although the wind turbine density is different for both wind farm clusters separated by the Kaskasi gap, no strong correlation was found between the density of the wind farms and the dominant scales of entrainment.

## 4 Discussion

### 4.1 Uncertainties

Since there were multiple flight legs for the four flights discussed in Sections 3.1 to 3.4, the data represented is the mean for all the flight legs at one location. The uncertainty in the auto-correlation, and turbulent momentum fluxes is represented by the Standard Error of Mean (SEM):

$$SEM = \frac{\sigma}{\sqrt{n}} \tag{9}$$

where $\sigma$ is the standard deviation and $n$ is the total number of flight legs at a location. Moreover, assuming that the flow
and atmospheric conditions are stationary during the whole duration of flight also brings uncertainty to the analysis since all the flight legs were not recorded at the same time. There are also some uncertainties arising from the conversion of wind vector from the geodetic coordinate system to the geographical coordinate system, and then transforming the horizontal wind component to the mean wind direction. These uncertainties arise from the systematic errors induced by pitch, roll, and yaw angle measurements, and usually, a correction factor is applied after the in-flight calibration procedure, as discussed by van den
Kroonenberg et al. (2008) and Lenschow (1986).

For the evaluation of turbulent momentum fluxes in Section 3.3, we utilized the rolling window of about 2 km to smooth the high-frequency data. The window length was chosen in order to reduce the random errors in first and second-order moments. Platis et al. (2018) recommended based on the work by Lenschow et al. (1994), that the rolling-window length should be more than 1800 m to include both small-scale variations in mean quantities and also incorporate large-scale flow effects. We tried
different rolling window lengths and observed their impact on the turbulent fluxes, and found the rolling window of 2 km to be the most representative of the flow phenomenon happening around these large wind farms.

Another important uncertainty in the analysis presented in Section 3.1 and Section 3.4 arises from the selection of the flight leg affected by the wake flow of wind farms. A two-tier strategy was applied to identify the wake and distinguish it from the large mesoscale effects. In the first step, the $u_t$ wind speed component minima for each leg were identified and a suitable
distance was added on both sides of the minima based on the method suggested by Cañadillas et al. (2020). The second step involved visual inspection of the portion of the flight leg identified as a wake flow in the first step. This was done in order to prevent a large mesoscale flow minimum to be identified as the wake flow. Especially during stable conditions, very long wakes were observed which experienced large-scale turning.





### 4.2 Regarding the turbulent momentum fluxes

From the variation in the lateral momentum flux $\langle u'v' \rangle$ discussed in Section 3.3, we observed that $\langle u'v' \rangle$ is predominantly negative during all the three flight legs in stable conditions. But the negative values of $\langle u'v' \rangle$ at the southern edge of wind farms (shown in Figure 9) negates the validity of flux-gradient hypothesis stated in Equation 10:

$$\langle u'v' \rangle = -K \frac{\partial u}{\partial y} \tag{10}$$

where $K$ is the eddy diffusivity constant. This relation implies that as the longitudinal wind component $u$ changes its magnitude in transverse direction ($y$-axis in Figure 1) due to the presence of wind farms, the $\partial u/\partial y$ gradient is negative at the Southern edge and positive at the Northern edge of these wind farms, which in turn should make $\langle u'v' \rangle$ positive at the Southern edge of wind farms and correspondingly negative for the Northern edge. The deviation from the flux-gradient relation can be explained by the fact that these measurements are not recorded in the surface layer but rather high up in the atmosphere and inside a canopy flow (for the flight legs above the wind farm). Above the surface layer or inside the canopy flow created by a wind farm, the momentum transport is dominated by large-scale eddies instead of the local gradient of wind or molecular diffusivity, and the validity of the flux-gradient relation can be questioned. This behavior has also been studied by Denmead and Bradley (1985) where they identified "counter-gradient fluxes" within a forest canopy flow because of large-scale turbulent transport eddies. We also observed an inverse correlation between $\langle u'v' \rangle$ and $\langle v'w' \rangle$ in all the flight recordings. For strong stable stratification cases, Pearson correlation coefficient values of $-0.85$, and $-0.72$ are recorded between $\langle u'v' \rangle$ and $\langle v'w' \rangle$ during above and upstream flight legs, respectively. While for the near-neutral stratification case, the correlation coefficient values are $-0.47$, and $-0.33$ during above and upstream flight legs, respectively.

We also observed that the case study wind farms did not satisfy the conditions of "infinite wind farm" because of a significant presence of lateral momentum flux $\langle u'v' \rangle$, especially during strong stable conditions. Under near-neutral conditions, the main source of energy transport inside the wind farms is the vertical flux of horizontal momentum $\langle u'w' \rangle$. By comparison, the lateral flux of momentum $\langle u'v' \rangle$ is quite low due to the weak gradient of $u$-component in the lateral direction. Under strong stable conditions, $\langle u'w' \rangle$ is still the main source of energy transport, but $\langle u'v' \rangle$ is also significant due to a sharp gradient of $u$-component in the lateral direction. This implies that in reality, large wind farms in offshore settings do not just rely on the vertical entrainment of momentum for energy recovery. Many analytical and engineering wake models for large offshore wind farms often ignore and exclude the lateral entrainment of momentum from the energy budget equation deeming it negligible (Emeis, 2022).

### 4.3 Regarding the length scales

From the analysis presented in Section 3.1, we observed that longitudinal length scales $\mathcal{L}_u$, and vertical length scales $\mathcal{L}_w$ manifest relatively different behavior under different stratification strengths. In the undisturbed flow, the difference between $\mathcal{L}_u$ at different strengths of stratification is not large: $\mathcal{L}_u$ in strong stable stratification is 2 to 3 times larger than the weak stratification (Figure 5). These large magnitudes of $\mathcal{L}_u$ represent mesoscale flow which is indicative of 2-D turbulence, comprising





extremely lower frequencies in the velocity spectrum. But the magnitude of $\mathcal{L}_w$ in the undisturbed flow strongly depends on the stratification strength in the atmosphere: $\mathcal{L}_w$ in strong stable stratification is about 10 to 15 times smaller than the weak or near-neutral stratification (see Figure 5). The analysis presented in Section 3.4 regarding the dominant scales of entrainment suggests that $\mathcal{L}_u$ does not influence the entrainment as much as $\mathcal{L}_w$. In the undisturbed flow, when $\mathcal{L}_u$ has large magnitudes

for both strong and weak stratification cases, we observed negligible momentum entrainment in the former case. Rather, the momentum entrainment is strongly correlated with the magnitude of $\mathcal{L}_w$ and the atmospheric stratification.

## 5  Conclusions

The flow structures inside and around large offshore wind farms strongly depend on atmospheric stability. In this study, we analyzed six different flight measurements around large offshore wind farms located in the North Sea to study the flow charac-
teristics based on the strength of observed thermal stratification. Under near-neutral stratification, large vertical length scales enhances mixing and instigates the wake recovery of large offshore wind farms. While in more stable conditions, mesoscale fluctuations in the transverse direction persist even in the wake flow causing less flow mixing and late wake recovery. Moreover, the rate at which energy is dissipated from large-scale motions to smaller turbulent structures also depends on the atmospheric stratification strength. The energy dissipation rate in the free atmosphere was about 40 times larger for neutral stratification
cases as compared to stable stratification. We also observed that the layout of the wind farm also influences the rate of energy dissipation and can also cause early or late wake recovery in the downstream flow.

Although the $\langle u'w' \rangle$ momentum flux is a major source of energy recovery in large wind farms, the case study wind farms did not conform with the "infinitely large wind farm" conditions. This is because of a significant presence of lateral momentum flux $\langle u'v' \rangle$, especially in strongly stable conditions. Under strongly stable conditions, there is negligible entrainment of
momentum flux $\langle u'w' \rangle$ in the undisturbed flow and the downstream wake flow of wind farms. The dominant length scales of entrainment range from 20 to $\sim 60$ m above the wind farm in all stratification strengths, and in the wake flow these scales range from 10 m to $\sim 100$ m only under near-neutral stratification. These scales are less than the rotor diameters of the wind turbines installed in the wind farms and provide much-needed vertical mixing to replenish the wake flow and increase power production in downstream wind farms. The $\langle u'w' \rangle$ entrainment length scales depicted a strong dependence on $\mathcal{L}_w$ rather than $\mathcal{L}_u$.
Understanding the turbulence length scales in offshore wind farms is imperative to the design of new farms, and to evaluating the power fluctuations and load intensities in large clusters of wind turbines.

*Data availability.* The airborne data set of the WIPAFF project is accessible to the community via the PANGAEA database https://doi.pangaea.de/10.1594/PANGAEA.902845



*Author contributions.* AHS and JM conceptualised and designed the study. AP and JB provided relevant data and analysis support. AHS
designed the objectives, performed analysis, and wrote the original draft manuscript. JM, AP, and JB supported the whole analysis and
reviewed and edited the whole manuscript.

*Competing interests.* JM is a member of the editorial board of Wind Energy Science.

*Acknowledgements.* This work is funded by the European Union Horizon 2020 research and innovation program under grant agreement
no. 861291 as part of the Train2Wind Marie Sklodowska-Curie Innovation Training Network (https://www.train2wind.eu/). The authors also
thank the crew of the WIPAFF campaign, Astrid Lampert, Rudolf Hankers, Thomas Feuerle, Mark Bitter and Helmut Schulz for their support.
The project WIPAFF was funded by the German Federal Ministry for Economic Affairs and Energy (Bundesministerium für Wirtschaft und
Energie) on the basis of a decision by the German Bundestag grant number: FKZ 0325783.



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
