# Peer review of "Turbulence structures and entrainment length scales in large offshore wind farms"

_Wind Energy Science, 2022_

## Author Response (AR1)

**Reviewers' comments**

**RC1**

The manuscript presents an analysis of mixing and turbulence in the wake of offshore wind farms. This is based on the WIPAFF campaign that took place the 2016-2017 winter. Relative to a number of simulations that have explored turbulence in the wakes of turbines, this study is based on a thorough analysis of dedicated airborne measurements. The analysis is sound, the measurements have great value for the wind energy sector and the understanding of the boundary layer in the vicinity of offshore wind farms. Some minor points and suggestions are made below (eg inclusion of error bars in figures providing estimates- of the length scales), but minor revision will be sufficient to address them and allow for publication.

Minor Concerns

l48 citation of Cortina: add parentheses

Authors' Response:

The parentheses are now added. (See l51)

l70-75: clearly stated objectives; not completely clear how objective 1 and 4 relate to each other: 1 aims at turbulent length scales, while 4 aims at dominant scales of entrainment.. both concern different locations relative to a wind farm; objective 4 gives more emphasis to the horizontal organization (around the wind farm, not just linearly upstream, above and downstream, and objective 4 includes a focus on different stratifications (objective 2). In other words, objective 4 seems to combine objectives 1 and 2 and to be somewhat redundant.

Authors' Response:

The authors have modified the description of the study objectives based on the reviewer's suggestion. (See l77)

Figure 1: it would be appropriate and useful to a concise description of the meteorological situation; at present, only the stability is discussed (figure 2..). The direction of the dominant wind should at least be indicated in figure 1, possibly

some isobars from reanalyses could be included to give a sense of the general direction of the flow.

**Authors' Response:**

**A new figure is now added which provides information about the dominant wind direction and prevalent stability conditions in the region of interest. (See Figure 1)**

l108-109: the vertical potential temperature gradient is indeed a measure of the stability of the atmosphere; it constitutes the basis for the calculation of the buoyancy frequency, or Brunt Vaisala frequency. This should be calculated and also given (either the buoyancy frequency or the period..), cf Holton, An introduction to dynamic meteorology (2004).

**Authors' Response:**

**The authors have calculated the buoyancy frequency for all the flights and included it in Table 1. (See Table 1)**

figure 5: the estimates include uncertainty; could error bars be added to give some appreciation of the uncertainty on these estimates? This would be hlpful to identify which features of the figure calls for interpretation, and which is likely without meaning.

**Authors' Response:**

**Error bars are now included in all the figures which indicate standard error of the mean values of the quantity being analyzed.**

l150: missing 'as': '...are more dominant, as clearly shown...'
**Authors' Response:**

**The word 'as' is now added. (See l165)**

This is not clear: the contrast between the black curves of the four panels is not so striking. The black curve of panel a extends to larger scales than that of panel c.

**There is only panel d (flight 40) which really sticks out as significantly different, with clearly larger scales present. (This is consistent with figure 5)**

**Authors' Response:**

For flights 32 and 33, the autocorrelation values drop quickly at lower spatial lags and then we get large values at large spatial lags because the atmosphere is slightly stable. For Flights 39 and 40, the autocorrelation values drop at a slower rate, and we observe a gradual decrease in the autocorrelation values. The main difference is the rate at which autocorrelation drops at lower spatial lags which correspond to the presence of small-scale structures in the flow.

**l170-180: The introduction and explanations for the formulas used for the spectra are insufficient or placed too late; a key quantity is wavenumber component ($k_2$) which is introduced before equation (4), but explained only after equation (5). References are given for more information on the subject, but there should be some more explanations given here nonetheless: what are the assumptions? What do these spectra correspond to?**
**Authors' Response:**

Assumptions of isotropic and incompressible flow are now included. The explanation of ($k_2$) is now given before its introduction in Equation 4. (See l190)

The spectra correspond to the inertial sub range and this has been mentioned in the text with underlying assumptions. (See l187 and l192)

**Regarding direction: $k_2$ is the wavenumber along the flight path because, by construction, this is the only spatial information we have access to with airborne measurements. The flight strategy implies that it is the wavenumber across the mean wind direction. Is this understanding right? The phrasing in line 183 puts the two information on the same level, which can create confusion.**
**Authors' Response:**

The phrasing has been changed to make it clearer to readers. The word 'across' has been changed to 'perpendicular'. (See l190)

**Figure 6: the choice is made to put the constant level (indicating dissipation rate) at the same height in the three panels; hence the reader must check the labels of the vertical axis to understand which is larger or weaker. It could be more visual**

and closer to expectations for such a figure to use the same vertical axis for the three panels. (Even if that leaves quite some space empty in the first panel..) However, the information is present in the following figure, so it may be fine to leave the figure as is.

Authors' Response:

The same vertical axis is now introduced in all three panels of this figure. As mentioned by the reviewer the information regarding the different magnitude of spectra in three locations is translated into the different magnitude of the rate of energy dissipation, which is shown in Figure 7.

l214: remove 'in': 'some momentum flux upstream..'
Authors' Response:

Done. (See l230)

l250-255: again, one needs to know for sure what $k_2$ corresponds to? Is it defined as the wavenumber along the profile (flight path)? Or as the wavenumber in the direction perpendicular to the main flow? (My understanding is that the flight paths were chosen perpendicular to the dominant wind, so that both coincidefor this specific campaign; it may be stated in section 2, but I missed it. It is worth emphasizing. A modification to Figure 1, including a description of the large-scale flow, would be welcome)
Authors' Response:

Yes, the flight paths were chosen perpendicular to the dominant wind. Here, $k_2$ corresponds to the wavenumber in the direction perpendicular to the main flow. It is now also mentioned in Section 3.2 where it is first introduced, and also described here in Section 3.4. (See l273)

l257: is the appropriate lengthscale to consider 1/k, or $2\pi / k$ ?

Authors' Response:

No consensus is present to define the length scale from wave number. Both definitions are present in the literature. Here the authors made a choice to define 1/k as a length scale.

**Figure 11: same remark as for Figure 1: helping the reader have an idea of the background flow for these newly introduced flows will be a significant improvement.**

**Authors' Response:**

This has been achieved with the addition of Figure 1 and a relevant explanation in the text. (See l93)

**Figure 12: error bars are necessary; uncertainties are discussed in section 4.1, a result of these considerations should be included in the figures, even in anticipation; this concerns Figure 12 and the previous figures which present estimates of scalars.**

**Authors' Response:**

Error bars are now added to all the figures where values from multiple flight legs were obtained at the same position. The error bars in these plots represent the standard error of the mean values of the quantity under discussion.

The only exception is Section 3.4.2 (Figure 13), where we only have single flight legs at every location downstream of the case study wind farms. This has been now mentioned in the text as well. (See l312)

**l360-361: the influence of farm layout has large practical implications. The authors should develop a bit more this statement: if some recommendation for farm layout may be tentatively stated, it is of interest; if the current study only hints at this influence, without making any conclusive statement possible, the authors could mention it and point to directions for further research.**

**Authors' Response:**

The authors have not made any conclusive statement on the effect of the wind farm layout on the energy entrainment inside the wind farm, although some results indicate that it could have important implications. This has been discussed and analyzed in many CFD/LES and numerical modeling studies but is not the main research question of this study. The authors' have made some recommendations on how to measure and analyze the effect of wind farm layout on kinetic energy

entrainment which can be an interesting research topic to pursue in the future. (See l360 and l403)

**l363: 'inifitely large wind farm' conditions: the authors should either explain or point to an appropriate reference (or both), both here and in the introduction.**

**Authors' Response:**

The authors have now added a brief description of the "infinite wind farm" case. Appropriate references are also added. (See l36)

**l370: providing perspectives for future research would be appropriate here; the study has the originality of using real observations, in contrast to many numerical studies. Is the way forward to carry out more detailed numerical simulations? Other observational campaigns (and then come other questions: what is missing from this one? Is it just necessary to sample more cases? Or should the measurements be different? Enhanced? More numerous, denser?)? Combinations of both?**

**Authors' Response:**

The authors have included their recommendations for the future work in the conclusions section. (See l403)
* * *
**RC2**

**This work presents turbulence mixing in the large offshore windfarms, based on the measurement done in WIPAFF campaign. The topic of this work is not directly my expertise. Therefore I cannot go into the details.**

**A good amount of analysis is done for this manuscript, which is valuable for the wind energy community. However, it seems the paper structure is not well defined. It seems there is not a line of the story to tell in the paper, which makes the paper hard to understand. The sections do not seem connected. In other words, it seems the paper is not coherent.  Also, there are some minor concerns:**

**l 33: needs a reference**

**Authors' Response:**

The reference has been added. (See l34)

**In the introduction section, more information about the measurement campaign can be useful.**

**Authors' Response:**

Several studies have already been published which contain detailed information about the WIPAFF measurement campaign. The studies are properly referenced in Section 1 and Section 2, and only the most relevant information is given in the text. (See l70 and l100)

**In section 2, the dominant wind direction plot can be useful.**

**Authors' Response:**

 A new figure is now added which provides information about the dominant wind direction and prevalent stability conditions in the region of interest. (See Figure 1)

**Section 3.4.1, the first paragraph, needs to be rewritten.**

**Authors' Response:**

The authors have modified the first paragraph of Section 3.4.1.

**The conclusion section needs to connect all 4 aspects of the analysis. That is missing.**

**Authors' Response:**

The authors have modified the conclusion based on the reviewers' suggestions.
* * *
**RC3**

This paper focuses on the variation in turbulence structures and dominant scales of vertical entrainment above large offshore wind farms located in the North sea based on the observation dataset of the research aircraft. This work is interesting as most of the previous research on entrainment is performed using large-eddy simulations layouts, which do not truly depict the reality. The analysis based on the actual in-situ measurements on real wind farms in this paper has good value for wind energy research and large offshore wind farms. Nevertheless, the structure should be improved as it is not easy to follow. It provided an extensive and detailed analysis, but the highlights of this article could be more clear.

I truly congratulate the authors for this manuscript, which I believe that should be considered for publication after minor changes.

The manuscripts, in my opinion, deserve revision on the following issues.

**Q1. l80-85 More information about the study area could be added in the text and Figure 1, such as the general location information (area, distance between land and sea), dominant wind direction, atmospheric circulation background, etc.**

**Authors' Response:**

A new figure is now added which provides information about the dominant wind direction and prevalent stability conditions in the region of interest. (See Figure 1)

This new information is also added to the text. (See l93)

**Q2. l260-270 Quantitative results need to be added to help make the results credible.**

**Authors' Response:**

The authors have already included the quantitative results in the plot and text. (See Figure 11)

**Q3. Discussion section needs to be improved.**

**Authors' Response:**

The authors have improved the Discussion section based on reviewers' suggestions.

**1) The highlight of this paper is the analysis based on observational data, rather than simulation experiments. Therefore, it is valuable to quantitatively compare the results of this study with those of previous studies based on simulation experiments.**

**2) The paper contains a lot of analysis, but it is not well connected. It is difficult to find the highlights and complete storyline. It should be improved in the discussion and conclusion parts.**

**Authors' Response:**

The authors have improved the Introduction, Discussion, and Conclusions sections based on the reviewers' suggestions.